# Alterations of the Skin and Gut Microbiome in Psoriasis and Psoriatic Arthritis

**DOI:** 10.3390/ijms22083998

**Published:** 2021-04-13

**Authors:** Irmina Olejniczak-Staruch, Magdalena Ciążyńska, Dorota Sobolewska-Sztychny, Joanna Narbutt, Małgorzata Skibińska, Aleksandra Lesiak

**Affiliations:** 1Department of Dermatology, Pediatric Dermatology and Dermatological Oncology, Medical University of Lodz, 91-347 Lodz, Poland; dorota.sobolewska-sztychny@umed.lodz.pl (D.S.-S.); joanna.narbutt@umed.lodz.pl (J.N.); malgorzata.skibinska@umed.lodz.pl (M.S.); aleksandra.lesiak@umed.lodz.pl (A.L.); 2Dermoklinika Centrum Medyczne, 90-436 Lodz, Poland; 3Department of Proliferative Diseases, Nicolaus Copernicus Multidisciplinary Centre for Oncology and Traumatology, 93-513 Lodz, Poland; ciazynska.magdalena@gmail.com

**Keywords:** psoriasis, psoriatic arthritis, microbiome, microbiota

## Abstract

Numerous scientific studies in recent years have shown significant skin and gut dysbiosis among patients with psoriasis. A significant decrease in microbiome alpha-diversity (abundance of different bacterial taxa measured in one sample) as well as beta-diversity (microbial diversity in different samples) was noted in psoriasis skin. It has been proven that the representation of *Cutibacterium*, *Burkholderia* spp., and *Lactobacilli* is decreased and *Corynebacterium kroppenstedii*, *Corynebacterium simulans*, *Neisseria* spp., and *Finegoldia* spp. increased in the psoriasis skin in comparison to healthy skin. Alterations in the gut microbiome in psoriasis are similar to those observed in patients with inflammatory bowel disease. In those two diseases, the *F. prausnitzii*, *Bifidobacterium* spp., *Lactobacillus* spp., *Parabacteroides* and *Coprobacillus* were underrepresented, while the abundance of *Salmonella* sp., *Campylobacter* sp., *Helicobacter* sp., *Escherichia coli*, *Alcaligenes* sp., and *Mycobacterium* sp. was increased. Several research studies provided evidence for the significant influence of psoriasis treatments on the skin and gut microbiome and a positive influence of orally administered probiotics on the course of this dermatosis. Further research is needed to determine the influence of the microbiome on the development of inflammatory skin diseases. The changes in microbiome under psoriasis treatment can serve as a potential biomarker of positive response to the administered therapy.

## 1. Introduction

Psoriasis is a chronic recurrent skin disease affecting approximately 2% of the global population. According to literature data, it is associated with higher prevalence of such comorbidities as cardiovascular disease, hypertension, obesity, metabolic syndrome and inflammatory bowel disease (IBD) [1]. Various drugs have been proposed in the treatment of this condition, such as topical medications based on corticosteroids and vitamin D derivatives, and systemic treatments based on suppressions of various inflammatory axes [2,3]. It has been shown that patients with psoriasis develop Crohn’s disease three times more often than the general population, while patients with Crohn’s disease will develop psoriasis seven times more often than the general population [4,5].

The influence of microbiota (i.e., the collection of bacteria, fungi, and parasites inhabiting the human body) has attracted the attention of scientists in recent years. Research on the microbiome—that is, the genetic material contained in these organisms—made it possible to learn more about the species composition in various parts of the human body, as well as to show changes in the course of various diseases.

## 2. Results

### 2.1. Microbiome Analysis Techniques

Research on the microbiome has used a variety of analysis techniques: cultured bacterial polymerase chain reaction (PCR), species-specific PCR, 16S rRNA (ribosomal ribonucleic acid) gene fragment, or full-length PCR, which hindered the possibility of an objective comparison of the test results [4]. However, the non-culture-dependent methods—DNA (deoxyribonucleic acid) and ribosomal RNA sequencing—resulted in the development of research on the microbiome. In the most recent studies on the microbiome, the sequencing of 16S rRNA V1–3 and V3–V4 regions, which is more effective and cheaper than metagenomic shotgun sequencing, was performed. In several papers, investigators indicated that V1–V3 amplicons give more informative data in cutaneous microbiome analysis, while V4 and V3–V4 are more appropriate in gut microbiome research [4]. The Human Microbiome Project, a huge project investigating the human microbiome with the use of 16S rRNA sequencing, enabled the dynamic development of further research on a specific ecosystem inhabiting the human body [6,7].

### 2.2. Alterations in Microbiome of the Skin in Psoriasis

According to literature data, bacterial phyla—*Actinobacteria*, *Bacteroidetes*, *Firmicutes* and *Proteobacteria*—and bacterial genera— *Cutibacterium*, *Corynebacterium* and *Staphylococcus*—predominantly inhabit healthy human skin [8]. The composition of the microbiota may vary significantly depending on the host’s place of residence, body area, age, presence of comorbidities, hygiene level, medications used, and external conditions [9].

Scientific studies in recent years have shown significant skin dysbiosis among psoriasis patients. A significant decrease in microbiome alpha-diversity (abundance of different bacterial taxa measured in one sample) as well as beta-diversity (microbial diversity in different samples) was noted in psoriasis skin [10]. A large cohort study was performed in order to analyze the microbiome of the skin swabs collected from 75 psoriasis patients (both lesional and non-lesional skin) and 124 controls [10,11]. Significant decreases in richness and diversity (expressed by the reduction of the Shannon index) were reported in lesional samples when compared to non-lesional and control samples [10,11].

In the observation by Fyhrquist et al. [12], the representation of *Cutibacterium*, *Burkholderia* spp., and *Lactobacilli* was decreased, while that of *Corynebacterium kroppenstedii*, *Corynebacterium simulans*, *Neisseria* spp., and *Finegoldia* spp. was increased in the psoriasis skin in comparison to healthy skin.

Literature data showed that *Corynebacterium* abundance was higher in more inflamed skin lesions [13]. Other study provided data on the positive correlation between *Staphylococcus* and *Corynebacterium* abundance and the Psoriasis Area Severity Index (PASI) score. *Corynebacterium* spp. can influence the interferon signaling pathway, which can lead to skin dysbiosis and the development of psoriatic lesions [6,10,12,13,14,15,16].

In the case of psoriatic skin lesions, scratching caused by itching may injure the skin; therefore, some bacteria (epidermal colonizers and other bacteria) can be found in the deep dermis or even in peripheral blood [16], where they easily encounter immune cells and cause congenital and adaptive inflammation [17,18,19,20]. This causes dysbiosis of the skin microbiota observed in the significantly reduced population of *Corynebacterium spp*., *Lactobacillus* spp., *Burkholderis* spp., and *Cutibacterium acnes* in psoriatic skin with lesions compared to healthy skin [18].

Non-culture techniques of skin microbiome analysis in psoriasis revealed a decreased abundance of *Burkholderia* spp., *Corynebacterium* spp., *Lactobacillus* spp. and *Cutibacterium*, with an increased generation of *Streptococcus* when compared to healthy controls [4,12,17]. Studies have shown inconsistent results on the abundance of *Staphylococcus* [9,16,21,22,23,24].

Literature data indicate that the skin affected by psoriatic lesions is characterized by an overrepresentation of *Firmicutes* phylum and a decreased population of *Actinobacteria* [16,17].

It has been proven that *Malassezia* is the abundant fungal genus of the human skin. Nevertheless, studies have shown that psoriatic lesions present higher diversity of fungal species than healthy skin [25]. The role of *Malassezia* in psoriasis pathogenesis is still unclear; however, according to Watanabe et al. [26], *M. sympodialis* can increase the production of the pro-inflammatory cytokines such as TNF-α (tumor necrosis factor alpha), IL-1 (interleukin 1), IL-6 (interleukin 6), and IL-8 (interleukin 8) in the skin and stimulate keratinocyte proliferation. Other studies revealed that this fungus may enhance the pro-inflammatory maturation of dendritic cells and the proliferation of mastocytes, which can stimulate the inflammation observed in psoriasis [27,28,29].

Significant clinical studies on the cutaneous microbiome in psoriasis are presented in Table 1.

### 2.3. The Role of Skin Microbiome in Psoriasis Pathogenesis

Interaction between the commensal organisms and the host occurs through recognition of the microbial-associated molecular patterns (MAMPs) by the specific pattern recognition receptors (PRRs) [32]. As a result of this interaction, microbiota can modulate the human postnatal immune system [33].

According to Fry et al. [34,35], there is a constant interaction between Toll-like receptors, peptidoglycan-recognition proteins, antimicrobial peptides, cytokines, and the microbiota of the human skin. It was proved that cathelicidins (LL-37), antimicrobial peptides, are produced by keratinocytes as a result of the contact with the commensal microorganisms. It binds with the nucleic acids of the epithelial cells, which were exposed as a result of apoptosis under external factors (e.g., bacterial, viruses, mechanical stress) in predisposed individuals. Self-DNA bonded to LL-37 stimulate the production of type I interferons by plasmacytoid dendritic cells (pDC), while self RNA interact with LL-37, which causes the production of TNFα and inducible nitric oxide synthase (iNOS) by myeloid dendritic cells (mDC) [33]. These cytokines affect the differentiation of the naive T cells into Th17 (T helper 17) cells, which produce interleukin IL-17 (interleukin 17) and IL-22 (interleukin 22), which lead to the development of psoriasis lesions [34].

Other research studies highlighted the role of *Candida albicans* in psoriasis pathogenesis. Dendritic cells activated by this fungus ligand, β-glucan, induce the production of IL-36α (interleukin 36 alpha), which leads to psoriasiform phenotype development [36].

Studies on a murine model of psoriasis showed that the animals treated with antibiotics, or held in a germ–free environment, did not develop chronic skin inflammation, and they showed a reduced formation of psoriasis-like plaques [37]. These reports may indicate the potential role of the microbiome in the development of inflammation in the skin and the formation of psoriasis.

What is worth mentioning is that the M protein of *Streptococcus pyogenes*, which colonizes the skin with psoriasis in a significant amount, exhibits molecular mimicry with the 50-kDa type I keratin. This activates autoreactive T cells and stimulates the inflammation, leading to the development of psoriasis [38,39].

The important role of the skin microbiome in psoriasis may be proved also by the fact that the concentration of beta defensin—an antimicrobial protein in the blood and skin of patients correlates with the concentration of IL-17, which is considered the main inflammatory cytokine in the pathogenesis of psoriasis. Moreover, it has been shown that the concentration of this protein decreases after treatment with secukinumab, which is an anti-IL-17 antibody and is directly proportional to PASI [40]. Further research is needed to investigate if the alteration of the skin microbiome in psoriasis is the cause or the consequence of the effective treatment of this dermatosis. 

### 2.4. Gut Dysbiosis in Psoriasis

Human intestinal microbiota constitutes the unique ecosystem consisting of more than 100 trillion cells, which are encoded by 5 million genes. It is composed of bacteria (mostly of six phyla: *Bacteroides*, *Actinobacteria*, *Fusobacteria*, *Firmicutes*, *Verrucomicrobia*, and *Proteobacteria*), fungi, viruses, protozoa and Archaea [41].

Alterations in the gut microbiome in psoriasis are similar to those observed in patients with inflammatory bowel disease and also in the individuals not diagnosed with IBD [42,43]. In those two diseases, the *F. prausnitzii*, *Bifidobacterium* spp., *Lactobacillus* spp., *Parabacteroide*, and *Coprobacillus* were underrepresented, while the abundance of *Salmonella* sp., *Campylobacter* sp., *Helicobacter* sp., *Escherichia coli*, *Alcaligenes* sp., and *Mycobacterium* sp. [42,43] was increased. The literature data indicate that the excessive inhabitancy of intestine by *Candida albicans*, *Malassezia* and *Staphylococcus aureus* can result in worsening of the psoriasis phenotype [43,44].

The gut microbiome in psoriasis is characterized by the increase in *Actinobacteria* and *Firmicutes* and also the *Firmicutes-to-Bacteroidetes* ratio (*F/B* ratio), which are the patterns of impaired gut epithelial barrier [45,46,47,48,49]. These lead to the stimulation of regulatory T cells, transport of carbohydrates and bacterial chemotaxis. On the other hand, *Ruminococcus* and *Megasphaera* are underrepresented in the microbiome in psoriasis.

Some studies revealed the increased abundance of Lachnospiraceae and Ruminococcaceae families, Collinsella aerofaciens, Dorea formicigenerans, and Ruminococcus gnavus species [50] and the underrepresentation of Faecalibacterium prausnitzii and Akkermansia muciniphila in the gut microbiome in psoriasis [50,51,52,53]. Faecalibacterium prausnitzii play an important role in the gut homeostasis by producing butyrates that have antioxidant properties, modulate the inflammatory response by the inhibition of the NF-κB (nuclear factor kappa light chain enhancer of activated B cells) and provide energy to the intestinal epithelial cells (enterocytes) [54,55,56]. These reports highlight the important role of the microbiome in maintaining a proper intestinal barrier.

Another study analyzing the gut microbiome in patients with psoriasis vulgaris and psoriatic arthritis revealed the decreased expression of the *Coprococcus* genus when compared to healthy controls. Moreover, in the patients with psoriatic arthritis, decreased abundance of the *Ruminococcus* and *Akkermansia* genera was noted. Authors suggest that it proves the progressive reduction in diversity accompanying the development of joint disease in patients with psoriasis [40]. Research studies showed a decreased concentration of medium-chain fatty acids (MCFA)—compounds that play a role in maintaining the integrity of the intestinal barrier [57]. *Actinobacteria* phylum was underrepresented in the microbiome of psoriatic arthritis patients [40]. On the other hand, the supplementation of Bifidobacterium species, which belong to the Actinobacteria, caused a decrease in the serum concentration of TNFα and C-reactive protein (CRP) in psoriasis patients.

According to the literature data, there is a strong association between increased skin and gut colonization by *Staphylococcus aureus*, *Candida albicans*, and *Malassezia* and psoriasis exacerbation [44].

Significant clinical studies on the intestinal microbiome in psoriasis are presented in Table 2.

### 2.5. Role of Gut Microbiome in Psoriasis Pathogenesis

Numerous studies have provided evidence for the existence of the gut–skin axis, which is dependent on the microbiome.

Okada et al. [63] revealed that the gut microbiome of the inflammatory skin murine model (keratinocyte-specific caspase-1 transgenic-Kcasp1Tg-mice) was characterized by an abundant population of *Staphylococcus aureus* and *Streptococcus danieliae*. The authors observed exacerbation of the skin lesions as well as an increased concentration of pro-inflammatory cytokines: IL-17A, IL-17F, IL-22 and TNF-α, after the administration of the *S. aureus* and *S. danieliae*. The presence of an impaired intestinal barrier has also been proven in the studies of Sikora et al. [64,65] in which elevated serum concentrations of intestinal fatty acid binding protein (I-FABP) and claudin-3 in the patients with psoriasis was observed. These reports also prove a significant role of intestinal dysbiosis in the development of psoriasis.

These findings may be proof for the existence of the vicious cycle: dermatitis inhabits specific gut bacteria that themselves worsen the inflammation of the skin.

It has been recently reported in the literature that as a result of the bacterial translocation process in patients with psoriasis, their DNA was presented in the blood of patients with active skin lesions. In addition, those DNA-positive individuals are characterized by an earlier onset, longer course of the disease, and higher concentrations of pro-inflammatory cytokines [66].

The changes in the microbiome associated with psoriasis can induce an inflammatory response by activating the cytokines IL-23, IL-17 and IL-22 as well as the modulation of the gamma interferon (IFN-γ) and inhibiting T regulatory cells (Treg) production. This leads to the uncalled growth of keratinocytes. A number of other inflammatory diseases are also associated with intestinal dysbiosis, e.g., IBD, Crohn’s disease. Interestingly, in patients with coexisting IBD and psoriasis, exacerbations of the skin lesions are accompanied by the exacerbations of the intestinal disease [48,59,67,68].

A reduced abundance of *Faecalibacterium prausnitzii* and *Akkermansia muciniphila* in psoriasis patients was elicited in the numerous studies [51]. It is worth mentioning that these bacteria produce short-chain fatty acids (SCFAs), which exhibit anti-inflammatory properties [69,70]. They have a positive effect on the functioning of the intestines by protecting against pathogenic microorganisms and preventing the development of intestinal inflammation [71]. Butyrate, the main SCFA, limits the production of reactive oxygen species, inhibits the adhesion, proliferation, translocation, and production of cytokines by cells of the immune response, and maintains a proper intestinal barrier [72,73]. Studies have shown that SCFAs also has the ability to inhibit the NF-κB signaling pathways-mediated response, block the production of IL-6 and thus reduce inflammation in the gut and other organs of the body [9,74,75].

According to scientific data, psoriasis, psoriatic arthritis, as well as the IBD and obesity, is connected with decreased abundance of an SCFA-producing bacteria: *Prevotella*, *Akkermansia*, *Faecalibacterium* and *Ruminococcus* [42,45,51]. On the basis of the mentioned studies, it can be concluded that intestinal dysbiosis in psoriasis and psoriatic arthritis is characterized primarily by a reduction in the occurrence of butyrate-producing bacteria. This phenomenon leads to the weakening of the intestinal barrier and disruption of the antigen presentation process as well as the translocation of bacteria from the intestinal lumen beyond its area and stimulation of the immune system, which results in the formation of a psoriatic phenotype [76,77]. Additionally, dysbiosis resulting in systemic inflammation, which includes the joints, is also a proposed model of psoriatic arthritis pathogenesis [59]. Several studies have reported a reduction in the concentration of enzymes involved in the synthesis of butyrates in fecal samples of patients with psoriasis [50]. Other reports indicated no difference in SCFA concentrations in the fecal samples of psoriasis patients compared to the healthy population while revealing a decreased abundance of SCFA-producing bacteria [42].

While SCFA exerts anti-inflammatory properties, it has been proven that MCFAs act opposite. They are underrepresented in the fecal samples of patients with psoriasis, psoriatic arthritis and IBD [42,78].

MCFAs show antibacterial properties, stimulate the peroxisome proliferator-activated receptor (PPAR), which results, among others, in a reduction of inflammation of the gut in patients with Crohn’s disease [79,80]. MCFAs stimulate the conversion of CD4 + lymphocytes to Th1 and Th17 cells, and they inhibit their conversion to Treg, while SCFAs act in the opposite way. Importantly, their influence on the differentiation of T lymphocytes after oral administration is mediated by intestinal microbes, which indicates a complex system of connections in the intestinal ecosystem [81]. Further research will provide a better understanding of the role of these particles in the development and course of inflammatory diseases, including psoriasis. They can influence the development of effective therapeutic strategies modifying the intestinal microbiome and thus influencing the course of autoimmune diseases [82].

There is evidence that the gut microbiota is capable of metabolizing tryptophan, which is an essential amino acid in protein synthesis. One of the substances formed as a result of this process is indole-3-aldehyde (IAld), which is involved in the local immune response through its effects on IL-22 and prevents the excessive proliferation of *Candida albicans* in the intestines. This phenomenon did not occur in mice reared in a microbial-free environment [83,84].

Several bacteria species have the ability to stimulate the production of regulatory T cells, which exhibit anti-inflammatory properties and are involved in maintaining immune tolerance. Moreover, these bacteria also have the ability to inhibit Th17, which is a key cell population involved in the development of the psoriasis [54,85]. Discussed by Eppinga et al., these findings suggest that *Faecalibacterium* and, likely, *Akkermansia* species are capable of negatively as well as systemically altering the immune system in the gut when present in deficient amounts [59]. Several studies have shown the anti-inflammatory effect of *Bifidobacterium*. In a murine model of colitis, Bifidobacterium bacteria induced the production of Treg [86]. In studies of patients with psoriasis, the supply of these bacteria resulted in a decrease in CRP and the pro-inflammatory cytokine TNFα in the serum of patients [87].

In psoriasis and psoriatic arthritis, not only disturbances in the intestinal microbiome but also alterations in the expression of the components of the immune response have been reported. Investigations of the patients’ fecal samples revealed an increased expression of soluble IgA (Immunoglobulin A) and decreased concentration of receptor activator of nuclear factor kappa-B ligand (RANKL) [42]. Interestingly, this protein, which is overexpressed in the serum and synovium of patients with psoriatic arthritis, acts as an osteoclastic activating factor, promoting the development of arthritis. However, in the intestines, it is responsible for the differentiation of *lamina propria* cells responsible for interaction with intestinal lumen antigens [42]. These variations in RANKL concentration may occur due to the specific effect of bacteria typical of psoriasis and psoriatic arthritis patients or indicate a modulating effect of this molecule on the development of systemic inflammation typical of psoriasis. 

Studies of fecal samples of patients with psoriasis, as well as inflammatory bowel diseases, showed an increased concentration of IL-1α, which is one of the key cytokines involved in the development of inflammation [88,89]. In psoriasis, this cytokine stimulates the accumulation of T lymphocytes and activates the process of antigen presentation and is also involved in the process of stimulating Th-17 lymphocytes in the skin of patients [90,91,92]. Therefore, increased IL-1α expression in the intestinal lumen of patients with psoriasis may be the link between the inflammation of the intestines accompanying this dermatosis and skin lesions [93,94].

The link between intestinal dysbiosis in psoriasis and the development of psoriatic comorbidities.

It has been proven that an elevated *F/B* ratio is often noted in psoriasis as well as in other conditions associated with systemic inflammation, such as cardiovascular disease, type-2 diabetes, and obesity [58,95,96,97]. Cho et al. noted that an elevated *F/B* ratio in the microbiota of healthy men is associated with increased amounts of trimethylamine-N-oxide (TMAO), which acts as a proatherogenic metabolite [95]. TMAO, by stimulating the macrophage activity and modifying the process of cholesterol metabolism, can lead to a higher risk of myocardial infarction, cardiovascular diseases, and stroke [98,99]. The initial process of TMAO production is the transformation of the carnitine derived from diet to trimethylamine (TMA), which is mediated by several bacteria. An increased prevalence of bacteria capable of metabolizing carnitine to TMA correlates with the *F/B* ratio and is associated with greater mortality and morbidity.

An increased *F/B* ratio may result in the limitation of SCFAs and MCFAs, as well as the butyrate production. Dysbiosis can also lead to increased acetate synthesis, which has been proven to be related to an excessive production of appetite stimulating hormone, ghrelin, and insulin resistance [100,101]. A reduction in butyrate concentration also predisposes the development of insulin resistance [72].

Studies on murine model have shown that the supply of butyrate to the diet of obese individuals resulted in the withdrawal of insulin resistance or prevented its development [102]. In obese people, an increased *F/B* ratio was also observed, and the decrease in *Firmicutes* concentration was directly proportional to the loss of body weight after introducing a low-calorie diet [103].

An increase in *Firmicutes* abundance was observed when the same individuals returned to their prior dietary habits and consequently gained weight [103]. Research indicates that obesity may be a result of intestinal dysbiosis, not just a consequence of it. Observations on mice demonstrated that the fecal transplantation from obese mice to germ-free individuals caused the weight gain of the latter [104]. Based on the literature data, it seems that a disturbed *F/B* ratio in patients with psoriasis may be a factor predisposing the development of diabetes and obesity [105,106].

On the basis of the above information, it seems that influencing the intestinal microflora of patients can improve not only the condition of the skin but also reduce the development of metabolic disorders.

### 2.6. The Role of Intestinal Dysbiosis in Psoriatic Arthritis 

Genome studies have provided evidence that polymorphisms in signaling pathways, including IL-23 (interleukin 23), may contribute to the development of inflammatory diseases such as spondyloarthropathies (SpA) and inflammatory bowel diseases [107,108]. Interesting results were provided by studies on the mouse model of arthritis and enteritis—SKG model, in which the carrier of the ZAP70 mutation is associated with disturbances in the signal pathway mediated by the prescription TCR (T-Cell Antigen Receptors), IL-23/IL-17 [109,110]. In this model, administration of the microbial component β1,3-glucan resulted in the development of enteritis and arthritis. However, the disease did not develop in individuals under conditions devoid of microbes [111]. Similarly, TNF-overexpressing mice (TNF∆ARE/+) used in scientific research as a model of development SpA and IBD did not develop ileitis under microbial-free conditions [112]. Recently, it has been shown that germ-free TNF∆ARE/+ mice do not develop Crohn-like ileitis [113].

The balance between the microbes inhabiting the intestine is possible due to the rich representation of cells modulating the immune response. Among them, we distinguish CD4+CD25+FOXP3+ Tregs, enabling the preservation of the immune tolerance state and Innate-like T cells (γδ T cells, mucosal-associated invariant T—MAIT, and invariant natural killer T cells—iNKT) [22,23,114]. Innate-like T cells show the ability to release a large number of cytokines (TNF, INF-γ, IL-4, IL-10, IL-17, and IL-22) as a result of activation of the TCR receptor but also in a mechanism independent of it.

iNKT cells have receptors (CD1d) that are sensitive to bacterial glycolipids, and MAIT cells are stimulated by the interaction of their receptors with metabolites of metabolism of riboflavin derived from bacteria and yeast [115]. These cells can exhibit both anti-inflammatory and pro-inflammatory effects under the action of IL-23 [112]. Their phenotype may also be changed under the influence of substances produced by intestinal bacteria as well as a result of interaction with Treg [112]. 

There are various hypotheses linking the development of spondyloarthropathies with disturbed intestinal microflora [113,114,115,116]. According to one of them, in childhood, an intestinal dysbiosis develops (which is related to the type of childbirth, the use of antibiotics at an early stage of life, or breastfeeding) [113,114,115,116]. Disturbances in the intestinal microbiome imply abnormal development and adaptation of the innate immune system, leading to chronic inflammation [117]. This hypothesis was reflected in observations of Praet et al. [118], who proved that mice lacking secondary lymphoid organs due to the lack of lymphotoxin were characterized by a specific type of intestinal dysbiosis. The disturbed bacterial microflora stimulated the production of IL-17 and the development of an autoimmune disease [118]. Another hypothesis recognizes a reversible relationship between genetic predisposition, the immune system, and the gut microbiome. Zanvit at al. [119] investigated that antibiotic therapy in adult mice resulted in a reduction in the severity of imiquimod-induced psoriasis-like dermatitis. Conversely, exposure to the same drugs early in life resulted in more severe course of inflammation (with elevated expression of IL-22-producing γδ (+) T cells) caused by imiquimod or recombinant IL-23 exposure in adulthood [119]. These observations may indicate an important role of microbes in the development of autoimmune diseases. 

Similar conclusions can be drawn based on reports that patients with ankylosing spondylitis were breastfed less frequently in infancy than the general population [120]. On the other hand, studies in rats have shown that the expression of HLA-B27 can significantly alter the intestinal metabolome [121].

Based on the literature data, interstitial dysbiosis can have a major impact on the psoriasis comorbidities development (Figure 1). 

### 2.7. Microbiome and Psoriasis Treatment

New research has provided evidence for the effects of psoriasis treatments on microbiome changes. Topical calcipotriol can inhibit the abundance of *Malassezia* population in the psoriatic skin by inducing the production of the antimicrobial peptide—cathelicidin [122,123].

Narrow-band ultraviolet radiation (NB-UVB) in psoriasis was proved to decrease the indicators of oxidative stress: malondialdehyde (MDA), reactive oxygen species (ROS), and ascorbyl radicals and cause the improvement in skin microbiome [124,125]. A study by Bosman et al. [126] showed that NB-UVB irradiation can (by influencing the metabolism of vitamin D) exert a significant influence on the intestinal microflora and confirms the existence of the skin–gut axis. Therefore, it can be presumed that the effectiveness of NB-UVB therapy in psoriasis patients may result not only from the direct influence of this radiation on the skin but also indirectly through the influence on the intestinal microbiome.

Langan et al. investigated the influence of various treatment approaches on the ratio of *Actinobacteria* to *Firmicutes*. From the analyzed methods of psoriatic treatment (conventional treatment: methotrexate, retinoids, cyclosporin A, and fumaric acid esters; conventional treatment plus phototherapy and biological treatment anti- TNFα, anti IL12/23 therapy), biological therapies had the strongest influence on the ratio of *Actinobacteria/Firmicutes* [127]. Scientists also observed a significant correlation of the Staphylococcus and Corynebacterium abundance PASI scores. 

In the other study, treatment with secukinumab—human anti-Th17 monoclonal antibody resulted in the higher expression of phylum *Proteobacteria* and lower abundance of *Firmicutes* and *Bacteroidetes* than in the therapy with ustekinumab (IL12/23 inhibitor) [128]. Moreover, a higher expression of *Enterobacteriacea* and *Pseudomonadaceae* under secukinumab therapy was observed. In contrast, no significant changes, except for genus *Coprococcus*, in the gut microbiome was observed under ustekinumab treatment. Significant differences in the baseline gut microbiome between responders and non-responders to secukinumab treatment were also noted. The authors concluded that the microbiome may be an important biomarker of the efficacy of secukinumab therapy [128].

### 2.8. Probiotics and Dietary Approaches in Psoriasis

Some scientific studies have provided evidence that the use of probiotics has a positive effect on the course of psoriasis [43,129,130].

*Lactobacillus pentosus* GMNL-77 and *Bifidobacterium infantis* 35,624 improved the psoriasis phenotype in an imiquimod-induced psoriasis murine model as well as in humans [87,131].

Decrease in plasma CRP and pro-inflammatory cytokines (i.e.,TNF-α and IL-6) has also been described, which can indicate a potential anti-inflammatory role of probiotics. The influence of probiotics on the immune system occurs probably due to the downregulation of CD103 + dendritic cells, which play a role in the antigen-presenting process, affecting the T regulatory cells in the human gut [132]. 

With these results in mind, it seems that the fecal transplant procedure, which is effectively used in the treatment of *Clostridium difficile* infection and inflammatory bowel disease [133,134,135], could be successfully applied in the treatment of inflammatory skin diseases such as psoriasis.

It has been demonstrated that the mean IgA anti-gliadin antibodies (AGA) are higher in patients with psoriasis than in the general population. Investigators evaluated the impact of a gluten-free diet in those who had with positive AGA tests on the severity of psoriasis [136,137]. In the group of IgA–AGA-positive patients on a gluten-free diet, a significant reduction in PASI was noted compared to the IgA–AGA-negative group. It is noteworthy that 60% of IgA–AGA positive patients experienced a deterioration in their skin condition after re-entering their normal diet. None of the IgA–AGA-negative patients noticed changes in their skin condition after returning to a regular diet [137]. Interestingly, the lower expression of the Ki-67+ cell population (a biomarker of cell proliferation) as well as dermal tissue transglutaminase in the psoriatic lesions after a gluten-free diet was noted. No significant changes in the skin of patients with a negative AGA test were discovered following the diet [138]. 

## 3. Materials and Methods

The PUBMED and Science Direct databases have been searched for articles relevant to this review. The analysis took into account the research studies published by March 2021. The databases were searched using the terms “microbiome” or “microbiota” and “psoriasis” or “psoriatic arthritis”. A total of 1204 search results were obtained. In the first phase of analysis, duplicated publications were eliminated (*=* 50). Subsequently, research studies wrote in any language other than English, performed on an animal model, meta-analyses, review articles, and case reports were excluded (*n* = 503). After the quality assessment, most relevant articles concerning alterations skin (Table 1) and gut (Table 2) microbiome in psoriasis and psoriatic arthritis were selected. The studies selection scheme is presented in the Figure 2.

## 4. Conclusions

Further research is needed to determine the influence of the microbiome on the development of inflammatory skin diseases. This would enable a better understanding of the pathogenesis of psoriasis and perhaps help to develop targeted therapies for this dermatosis. The changes in microbiome under psoriasis treatment can serve as a potential biomarker of positive response to the administered therapy.

## Figures and Tables

**Figure 1 ijms-22-03998-f001:**
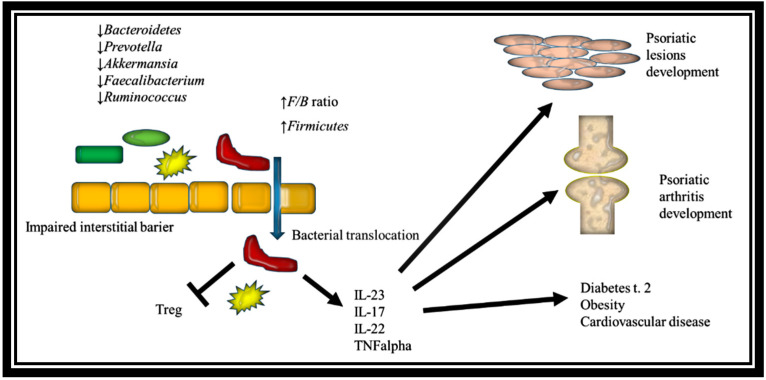
The impact of gut dysbiosis in psoriasis on the development of psoriatic lesions, psoriatic arthritis, and psoriasis comorbidities (*F/B* ratio - *Firmicutes-to-Bacteroidetes* ratio; Treg - T regulatory cells; TNFalpha - tumor necrosis factor alpha).

**Figure 2 ijms-22-03998-f002:**
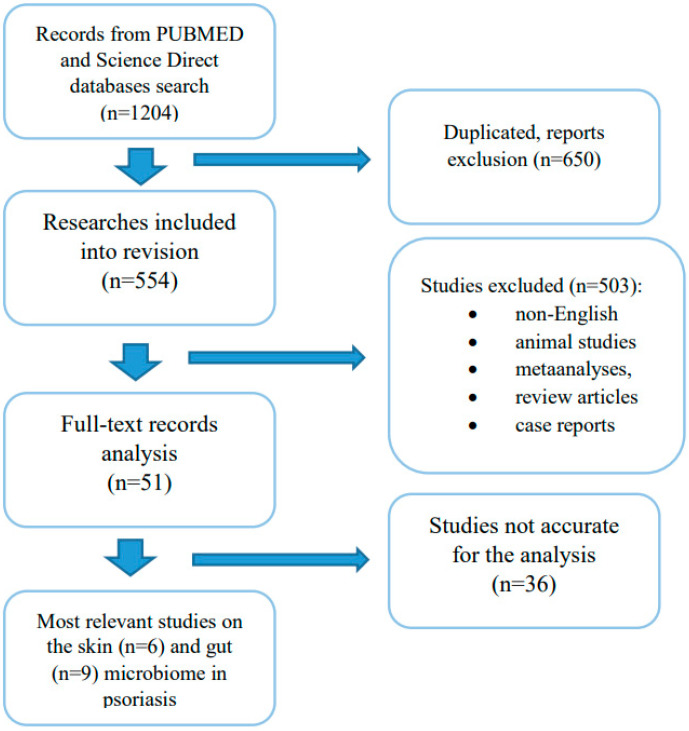
A graphical diagram of the selection of the literature data for the review.

**Table 1 ijms-22-03998-t001:** Summary of the most relevant research studies on the skin microbiome of patients with psoriasis.

Author	Study Group	Analysed Sample	Method of Analysis	Results
Fahlen et al., 2012 [17]	Psoriasis patients (*n* = 10) Healthy controls (*n* = 12)	Skin biopsy	16S rRNA sequesting (V3–V4 hypervariable region)	↑ *Proteobacteria* phylum ↓ *Firmicutes* and *Actinobacteria* phyla ↓ *Streptococci* and *Cutibacterium* genera ↑ *Staphylococci* genera
Alekseyenko et al., 2013 [11]	Psoriasis patients (*n* = 54) Healthy controls (*n* = 37)	Skin swab	16S rRNA sequesting (V1–V3 hypervariable region)	↓ *Proteobacteria* phylum ↑ *Actinobacteria* and *Firmicutes* phyla
Chang et al., 2018 [30]	Psoriasis patients (*n* = 28) Healthy controls (*n* = 26)	Skin swab	16S rRNA sequesting (V1–V3 hypervariable region)	↓ *Actinobacteria* phylum ↑ *Proteobacteria* phylum ↓ *Cutibacterium*, *Ethanoligenens* and *Macrococcus* genera ↑ *Pseudomonas* genera ↓ *Cutibacterium acnes*, *Cutibacterium granulosum*, *Staphylococcus epidermidis* ↑ *Staphylococcus aureus* and *Staphylococcus pettenkoferi*
Fyhrquist et al., 2019 [12]	Atopic dermatitis patients (*n* = 82) Psoriasis patients (*n* = 119) Healthy controls (*n* = 115)	Skin swab	16S rRNA sequesting (V1-V4 hypervariable region)	↑ *Finegoldia*, *Neisseriaceae*, *Corynebacterium kroppenstedtii*, *Corynebacterium simulans* ↓ *Lactobacilli*, *Burkholderia* spp., *Cutibacterium acnes*
Assarsson et al., 2020 [31]	Psoriasis patients (*n* = 50) Healthy controls (*n* = 77)	Skin swab	16S rRNA sequesting (V3-V4 hypervariable region)	↑ *Firmicutes* and *Protebacteria* phyla ↓ *Fusobacteria* and *Cyanobacteria* phyla
Quan et al., 2020 [13]	Psoriasis patients (*n* = 27) Healthy controls (*n* = 19)	Skin swab	16S rRNA sequesting (V3–V4 hypervariable region)	↓ *Deinococcus* and *Thermus* phyla ↑ *Corynebacterium* genera ↓ *Cutibacterium* genera

↓—decreased; ↑—increased; rRNA—ribosomal ribonucleic acid.

**Table 2 ijms-22-03998-t002:** Summary of the most relevant research studies on the gut microbiome of patients with psoriasis.

Author	Study Group	Analysed Sample	Method of Analysis	Results
Scher et al., 2015 [42]	Psoriasis patients (*n* = 15) Psoriatic arthritis patients (*n* = 16) Healthy controls (*n* = 17)	Fecal sample	16s rRNA sequencing (V1 -V2 hypervariable region)	↓ diversity in psoriasis and psoriatic arthritis ↓ *Bacteroidetes* phylum in psoriasis versus psoriatic arthritis ↓ *Akkermansia*, *Alistipes*, *Parabacteroides*, *Pseudobutyrivibrio*, *Ruminococcus* and *Coprococcus* genera in psoriatic arthritis and psoriasis ↓ *Coprobacillus* genera in psoriasis when compared to psoriatic arthritis
Masallat et al., 2016 [58]	Psoriasis patients (*n* = 45) Healthy controls (*n* = 45)	Fecal sample	Real-time PCR	↓ *Actinobacteria* phylum ↑ *Bifidobacterium Collinsella*, *Dorea Ruminococcus*, *Slackia* and *Subdoligranulum* genera ↑ F/B ratio
Eppinga et al., 2016 [59]	Psoriasis patients (*n* = 29) IBD patients (*n* = 31) HS patients (*n* = 17)-Psoriasis and IBD patients (*n* = 13) HS and IBD patients (*n* = 17) Healthy controls (*n* = 33)	Fecal sample	Quantitative PCR	↓ *F. prausnitzii* ↑ *E. coli* in patients with psoriasis and concomitant psoriasis and IBD
Tan et al., 2018 [51]	Psoriasis patients (*n* = 14) Healthy controls (*n* = 14)	Fecal sample	16s rDNA sequencing (V4 hypervariable region)	↓ *Akkermansia muciniphila*, *Verrucomicrobia* and *Tenericutes* phyla *Mollicutes* and *Verrucomicrobiae* ↑ *Bacteroides* genera, *Clostridium citroniae* spp. and *Enterococcus* genera
Chen et al., 2018 [45]	Psoriasis patients (*n* = 32) Non-psoriasis controls (*n* = 64)	Fecal sample	16s rRNA sequencing (V3 - V4 hypervariable region)	↑ *Firmicutes* ↓ *Bacteroidetes* ↑ *F/B* ratio ↓ *Akkermansia* genus
Hidalgo -Cantabrana et al., 2019 [60]	Psoriasis patients (*n* = 19) Healthy controls (*n* = 20)	Fecal sample	16s rRNA sequencing (V2 -V3 hypervariable region)	↓ diversity ↑ *Firmicutes* ↓ *Bacteroidetes* ↑ *F/B* ratio ↑ *Actinobacteria* ↓ *Proteobacteria* phylum, *Alistipes*, *Bacteroides*, *Barnesiella*, *Faecalibacterium*, *Parabacteroides* and *Paraprevotella* genera
Shapiro et al., 2019 [50]	Psoriasis patients (*n* = 24) Non-psoriasis controls (*n* = 24)	Fecal sample	16s rRNA sequencing (V4 hypervariable region)	↑ *Firmicutes ↓ Bacteroidetes* ↑ *F/B* ratio ↑ *Actinobacteria* phylum, *Blautia* and *Faecalibacterium* genera ↓ *Proteobacteria* phylum, *Prevotella genarum*, *Ruminococcus gnavus*, *Dorea formicigenerans* and *Collinsella aerofaciens* spp.
Zhang et al., 2021 [61]	Psoriasis patients (*n* = 30) Healthy controls (*n* = 30)	Fecal sample	16s rRNA sequencing	↑ *Faecalibacterium* and *Megamonas* taxa
Valentini et al., 2021 [62]	Psoriasis patients treated with biologic therapy (*n* = 10) Psoriasis patients not treated with biologic therapy (*n* = 20)	Fecal sample	16s rRNA sequencing	↓ diversity of biologically treated patients vs. untreated patients

↓—decreased; ↑—increased; rRNA—ribosomal ribonucleic acid.

## Data Availability

No new data were created or analyzed in this study. Data sharing is not applicable to this article.

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
