# Peer review of "Alterations of the Skin and Gut Microbiome in Psoriasis and Psoriatic Arthritis"

_ijms, 2021, doi:10.3390/ijms22083998_

Round 1

Reviewer 1 Report

A very comphrehensive review describing the role of cutaneous and intestinal microbiome in psoriasis, focusing specifically in artropatic psoriasis; I have some queries.

A matherials and methos subsection, stating what databases were searched in order to perform this review, if there were any preferential criteria in study selection, would in my opinion be a great add to the study; 

Page 2 line 31 this senntence needs a referral: "Psoriasis is a chronic recurrent skin disease affecting approximately 2% of the global population. According to literature data, it is associated with higher prevalence of such comorbidities as cardiovascular disease, hypertension, obesity, metabolic syndrome and inflammatory bowel disease (IBD)." I suggest you this one :doi: 10.1371/journal.pone.0241575.

Page 2 line 33-34, you should add: "Various drugs have been proposed in the treatment of this condition, such as topical medications based on corticosteroids and vitamin D derivatives , and systemic treatments based on suppressions of various inflammatory axis." and cite an aryticle such as:  doi: 10.1111/dth.13185.and doi: 10.1111/dth.14504. 

Thank You

Author Response

Dear Reviewer,     

Thank You for Your valuable comments. According to them the section “2. Material and Methods” has been added to the manuscript.

The specified references and sentence has also been added to manuscript.

Kind regards,

Irmina Olejniczak-Staruch

Reviewer 2 Report

The work is a complete revision on studies regarding skin and gut microbiome in psoriasis and psoriatic arthritis.

I found the paper very complete

I think that a flow chart of study selection would be a great addition to the study;

Author Response

Dear Reviewer 2,

Thank You for your opinion. According to your suggestions the flow chart of study selection has been added to the manuscript within the new section "2. Material and Methods”.

Kind regards,

Irmina Olejniczak-Staruch

Round 2

Reviewer 1 Report

The author responded to all queries. the paper is ready to be published.